# Uniqueness of Single Peak Solutions for a Kirchhoff Equation

Junhao Lv [1], Shichao Yi [1,2,*] and Bo Sun [1]

1   School of Science, Jiangsu University of Science and Technology, Zhenjiang 212003, China;
    212210501118@stu.just.edu.cn (J.L.); sunbohit@163.com (B.S.)
2   Yangzijiang Shipbuilding Group, Taizhou 212299, China
*   Correspondence: shichaoyi@just.edu.cn

**Abstract:** We deal with the following singular perturbation Kirchhoff equation: $-\left(\epsilon^2 a + \epsilon b \int_{\mathbb{R}^3} |\nabla u|^2 dy\right) \Delta u + Q(y)u = |u|^{p-1}u$, $u \in H^1(\mathbb{R}^3)$, where constants $a, b, \epsilon > 0$ and $1 < p < 5$. In this paper, we prove the uniqueness of the concentrated solutions under some suitable assumptions on asymptotic behaviors of $Q(y)$ and its first derivatives by using a type of Pohozaev identity for a small enough $\epsilon$. To some extent, our result exhibits a new phenomenon for a kind of $Q(x)$ which allows for different orders in different directions.

**Keywords:** Kirchhoff equations; single-peak solutions; uniqueness; Pohozaev identity

**MSC:** 35A01; 35A02; 35B25; 35J20; 35J60

## 1. Introduction

In 1746, D'Alembert first formulated the wave equation in his treatise and proved its functional relationships in 1750. The study of elastic string vibrations pioneered the discipline of partial differential equations. In 1883, Kirchhoff [1] extended the classical D'Alembert wave equation to the free vibration of elastic strings by considering a physical model for the change in string length due to transverse vibrations.

$$\rho \frac{\partial^2 u}{\partial t^2} - \left(\frac{P_0}{h} + \frac{E}{2L} \int_0^L \left|\frac{\partial^2 u}{\partial y^2}\right| dy\right) \frac{\partial^2 u}{\partial y^2} = 0, \tag{1}$$

where $L$ is the length of the string, $h$ is the cross-sectional area, $E$ is the Young's modulus of the material, $\rho$ is the mass density, and $P_0$ is the initial tension. With further research, scholars have found that Kirchhoff-type equations have a wealth of applications [2–4] and have become a typical class of issues in partial differential equations.

In this paper, we are concerned with the following nonlocal Kirchhoff problem

$$-\left(\epsilon^2 a + \epsilon b \int_{\mathbb{R}^3} |\nabla u|^2 dy\right) \Delta u + Q(y)u = |u|^{p-1}u, \ u \in H^1(\mathbb{R}^3), \tag{2}$$

where $\epsilon > 0$ is a small parameter, and constants $a, b > 0$ and $1 < p < 5$.

In recent decades, there has been considerable interest in the existence and uniqueness of solutions for (2) under suitable conditions on the function $Q(y)$. In particular, when $\epsilon = 1$ and $Q(y)$ is a constant, the existence and non-degeneracy of ground state solutions were implied in [5,6]. Using the non-degeneracy of ground states, in [5], Li et al. added the existence and uniqueness of single-peak solutions to (2) and Luo, Peng, Wang and Xiang [7] obtained the existence of multi-peak positive solutions of (2) by combining the variational method and the Lyapunov–Schmidt reduction for small $\epsilon$. For more works concerning the uniqueness of concentrated solutions, one can refer to [8–12].

Now, we state the conditions of $Q(y)$ in [5] as follows:

**($Q_1$)** $Q(y)$ is a bounded $C^1$ function with $\inf\limits_{y \in \mathbb{R}^3} Q(y) > 0$.

**(Q₂)** *There exist $y^0 \in \mathbb{R}^3$ and $r_0 > 0$ such that $Q(y^0) < Q(y)$ for $0 < |y - y^0| < r_0$.*
**(Q₃)** *There exist $m > 1$ and $\rho > 0$ such that*

$$\begin{cases} Q(y) = Q(y^0) + \sum_{j=1}^3 c_j |y_j - y_j^0|^m + O(|y - y^0|^{m+1}), \ y \in B_\rho(y^0), \\ \frac{\partial Q(y)}{\partial y_j} = mc_j |y_j - y_j^0|^{m-2}(y_j - y_j^0) + O(|y - y^0|^m), \ y \in B_\rho(y^0), \end{cases}$$

*where $\rho > 0$ is a small constant and $c_j \neq 0$ for $j = 1, 2, 3$.*

**Theorem 1** (c.f. [5]). *Suppose that $Q(y)$ satisfies $(Q_1)$, $(Q_2)$ and $(Q_3)$. Let $u_\epsilon^{(i)}, i = 1, 2$ be two positive solutions of (2) concentrating at the same point $y^0$. Then, $u_\epsilon^{(1)} = u_\epsilon^{(2)}$ for a sufficiently small $\epsilon$.*

Here, we want to mention that the authors in [5] used the assumption that $Q(y)$ has the same order in different directions at $y^0$. However, to our knowledge, whether there is similar uniqueness when $Q(y)$ has different increasing rates in different directions is still unknown. In this paper, we give an answer on this aspect and we consider a class of $Q(y)$ as follows:
**(Q̄₃)** *$Q(y^0) < Q(y)$ for any $y \in \mathbb{R}^3 \setminus \{y^0\}$ and $Q(y)$ satisfies*

$$\begin{cases} Q(y) = Q(y^0) + \sum_{j=1}^3 c_j |y_j - y_j^0|^{m_j} + O(|y - y^0|^{m+1}), \ y \in B_\rho(y^0), \\ \frac{\partial Q(y)}{\partial y_j} = mc_j |y_j - y_j^0|^{m_j-2}(y_j - y_j^0) + O(|y - y^0|^m), \ y \in B_\rho(y^0), \end{cases}$$

*where $\rho > 0$ is a small constant, $m_j > 1$, $m = \max\{m_1, m_2, m_3\}$ and $c_j \neq 0$ for $j = 1, 2, 3$.*

**Theorem 2.** *Suppose that $Q(y)$ satisfies $(Q_1)$ and $(\bar{Q}_3)$. Then, (2) has only one positive single-peak solution if $\epsilon$ is small enough.*

## 2. Some Basic Estimates

Let $U_{y^0}(y)$ be the unique positive solution of the following problem:

$$\begin{cases} -\left(a + b \int_{\mathbb{R}^3} |\nabla u|^2 dy\right)\Delta u + Q(y^0)u = |u|^{p-1}u, \text{ in } \mathbb{R}^3, \\ u(0) = \max_{y \in \mathbb{R}^3} u(y), \ u(y) \in H^1(\mathbb{R}^3). \end{cases}$$

It follows from [5] that $U_{y^0}(y)$ is a radially symmetric decreasing function satisfying

$$|D^\alpha U_{y^0}(y)| \leq Ce^{-\delta|y|}, \text{ with } |\alpha| \leq 1 \text{ and some } C, \delta > 0.$$

First we denote

$$\|u\|_\epsilon = (u(y), u(y))_\epsilon^{\frac{1}{2}} = \left(\int_{\mathbb{R}^3} (\epsilon^2 a|\nabla u|^2 + Q(y)u^2(y))\right)^{\frac{1}{2}},$$

and for $x \in \mathbb{R}^3$, we let

$$E_{\epsilon,x} = \left\{u \in H^1(\mathbb{R}^3) : (u(y), U_{y^0}(\tfrac{y-x}{\epsilon}))_\epsilon = 0, \ (u(y), \frac{\partial U_{y^0}(\frac{y-x}{\epsilon})}{\partial y_j})_\epsilon = 0, j = 1, 2, 3\right\}.$$

By using the standard Lyapunov–Schmidt reduction as that in Theorem 1.3 in [5], the following basic structure of the concentrated solutions can be obtained.

**Proposition 1.** *Suppose that $Q(y)$ satisfies $(Q_1)$ and $(\bar{Q}_3)$. Then, there exists $\epsilon_0$ such that for all $\epsilon \in (0, \epsilon_0)$, problem (2) has a solution $u_\epsilon$ of the form*

$$u_\epsilon(y) = U_{y^0}(\frac{y - y_\epsilon}{\epsilon}) + \omega_\epsilon(y), \tag{3}$$

*with $y_\epsilon, \omega_\epsilon \in E_{\epsilon, y_\epsilon}$ satisfying*

$$|y_\epsilon - y^0| = o(1), \quad \|\omega_\epsilon\|_\epsilon = o(\epsilon^{\frac{3}{2}}). \tag{4}$$

Now, we consider

$$L_\epsilon(\omega_\epsilon) = -\left(\epsilon^2 a + \epsilon b \int_{\mathbb{R}^3} |\nabla U_{y^0}(\frac{y - y_\epsilon}{\epsilon})|^2\right) \Delta \omega_\epsilon + 2\epsilon b \left(\int_{\mathbb{R}^3} \nabla U_{y^0}(\frac{y - y_\epsilon}{\epsilon}) \nabla \omega_\epsilon\right) \Delta U_{y^0}(\frac{y - y_\epsilon}{\epsilon})$$
$$+ Q(y)\omega_\epsilon - p U_{y^0}^{p-1}(\frac{y - y_\epsilon}{\epsilon})\omega_\epsilon.$$

We can rewrite $L_\epsilon(\omega_\epsilon)$ as

$$L_\epsilon(\omega_\epsilon) = R_\epsilon(\omega_\epsilon) + N_\epsilon(\omega_\epsilon), \tag{5}$$

where $R_\epsilon(\omega_\epsilon) = (Q(y^0) - Q(y)) U_{y^0}(\frac{y - y_\epsilon}{\epsilon})$, and

$$\begin{aligned} N_\epsilon(\omega_\epsilon) = &\{2\epsilon b \left(\int_{\mathbb{R}^3} \nabla U_{y^0}(\frac{y - y_\epsilon}{\epsilon}) \nabla \omega_\epsilon\right) \Delta U_{y^0}(\frac{y - y_\epsilon}{\epsilon}) \\ &+ \epsilon b \int_{\mathbb{R}^3} \left(2\nabla U_{y^0}(\frac{y - y_\epsilon}{\epsilon}) \nabla \omega_\epsilon + |\nabla \omega_\epsilon|^2\right) \Delta (U_{y^0}(\frac{y - y_\epsilon}{\epsilon}) + \omega_\epsilon)\} \\ &+ \{(U_{y^0}(\frac{y - y_\epsilon}{\epsilon}) + \omega_\epsilon)^p - U_{y^0}^p(\frac{y - y_\epsilon}{\epsilon}) - p U_{y^0}^{p-1}(\frac{y - y_\epsilon}{\epsilon})\omega_\epsilon\} \\ =: &N_\epsilon^2(\omega_\epsilon) + N_\epsilon^1(\omega_\epsilon). \end{aligned}$$

**Lemma 1** (c.f. [5]). *There exist $\epsilon_1 > 0, \rho_1 > 0$ and $\gamma > 0$ sufficiently small such that for any $\epsilon \in (0, \epsilon_1), \rho \in (0, \rho_1)$,*

$$|\int_{\mathbb{R}^3} L_\epsilon(\omega_\epsilon)\omega_\epsilon| \geq \gamma \|\omega_\epsilon\|_\epsilon^2$$

*holds uniformly with respect to $y_\epsilon \in B_\rho(y^0)$.*

**Proposition 2.** *It holds*

$$\|\omega_\epsilon\|_\epsilon = O(\epsilon^{\frac{3}{2} + \min\{m_1, m_2, m_3\}}) + O(\epsilon^{\frac{3}{2}} \max_{j=1,2,3} |y_{\epsilon,j} - y_j^0|^{m_j}).$$

**Proof.** First, using the condition $(\bar{Q}_3)$ and the Hölder inequality, for a small constant $d$, we have

$$\begin{aligned} |\int_{B_d(y_\epsilon)} R_\epsilon(\omega_\epsilon)\omega_\epsilon| &= |\int_{B_d(y_\epsilon)} (Q(y^0) - Q(y)) U_{y^0}(\frac{y - y_\epsilon}{\epsilon})\omega_\epsilon| \\ &\leq C \sum_{j=1}^3 \int_{B_d(y_\epsilon)} |y_j - y_j^0|^{m_j} U_{y^0}(\frac{y - y_\epsilon}{\epsilon}) |\omega_\epsilon| \\ &\leq C\epsilon^{\frac{3}{2}} \sum_{j=1}^3 (\epsilon^{m_j} + |y_{\epsilon,j} - y_j^0|^{m_j}) \|\omega_\epsilon\|_\epsilon, \end{aligned} \tag{6}$$

where $y_j, y_{\epsilon,j}, y_j^0$ denote the $j$th components of $y, y_\epsilon, y^0$.

Moreover, by the exponential decay of $U_{y^0}(\frac{y - y_\epsilon}{\epsilon})$, we can obtain that for any $\sigma > 0$,

$$|\int_{\mathbb{R}^3 \setminus B_d(y_\epsilon)} R_\epsilon(\omega_\epsilon)\omega_\epsilon| \leq C\epsilon^\sigma \|\omega_\epsilon\|_\epsilon. \tag{7}$$

Thus, (6) and (7) give that

$$|\int_{\mathbb{R}^3} R_\epsilon(\omega_\epsilon)\omega_\epsilon| = O(\epsilon^{\frac{3}{2} + \min\{m_1, m_2, m_3\}}) \|\omega_\epsilon\|_\epsilon + O(\epsilon^{\frac{3}{2}} \max_{j=1,2,3} |y_{\epsilon,j} - y_j^0|^{m_j}) \|\omega_\epsilon\|_\epsilon. \tag{8}$$

On the other hand, it can be directly calculated that

$$\left| \int_{\mathbb{R}^3} N_\epsilon^1(\omega_\epsilon)\omega_\epsilon \right| \leq C \int_{\mathbb{R}^3} |\omega_\epsilon(y)|^{\min\{p+1,3\}} = o(1)\|\omega_\epsilon\|_\epsilon^2, \tag{9}$$

$$\left| \int_{B_d(y_\epsilon)} N_\epsilon^2(\omega_\epsilon)\omega_\epsilon \right| = \left| -3\epsilon b \int_{\mathbb{R}^3} \nabla U_{y^0}\left(\frac{y-y_\epsilon}{\epsilon}\right) \nabla \omega_\epsilon(y) \int_{\mathbb{R}^3} |\nabla \omega_\epsilon|^2 - \epsilon b\left(\int_{\mathbb{R}^3} |\nabla \omega_\epsilon|^2\right)^4 \right|$$
$$= o(1)\|\omega_\epsilon\|_\epsilon^2. \tag{10}$$

So, from (5), (8)–(10) and Lemma 1, the result follows. □

**Proposition 3.** *Suppose that $u_\epsilon(y)$ is a positive solution of (2). Then, for any $R \gg 1$, there exist $\eta > 0$ and $C > 0$ such that*

$$|u_\epsilon(y)| + |\nabla u_\epsilon(y)| \leq Ce^{-\eta \frac{|y-y_\epsilon|}{\epsilon}}, \quad x \in \mathbb{R}^3 \backslash B_{R\epsilon}(y_\epsilon). \tag{11}$$

**Proof.** Using the comparison principle of He and Xiang [13], we can obtain (11), which also can be found in [5]. □

Let $u(y)$ be a positive solution of (2). Then, by multiplying $\partial_{y_j} u$ on both sides of (2) and then integrating by parts, we have for each $j = 1, 2, 3$

$$\int_{\mathbb{R}^3} \frac{\partial Q}{\partial y_j} u^2(y) dy = 0. \tag{12}$$

**Proposition 4.** *Let $u_\epsilon(y)$ be the solution of (2) with the form (3) and (4). Assume that $(Q_1)$ and $(\bar{Q}_3)$ hold. Then,*

$$\|\omega_\epsilon\|_\epsilon = O(\epsilon^{\frac{3}{2} + \min\{m_1, m_2, m_3\}}) \quad and \quad |y_\epsilon - y^0| = o(\epsilon).$$

**Proof.** First, (11) and (12) tell us that for a small $d > 0$, there exists some $\sigma > 0$ such that

$$\int_{B_d(y_\epsilon)} \frac{\partial Q}{\partial y_j} \left( U_{y^0}\left(\frac{y-y_\epsilon}{\epsilon}\right) + \omega_\epsilon \right)^2 dy = O(e^{-\frac{\sigma}{\epsilon}}).$$

Also, similar to (6), we have

$$\left| \int_{B_d(y_\epsilon)} \frac{\partial Q}{\partial y_j} U_{y^0}\left(\frac{y-y_\epsilon}{\epsilon}\right) \omega_\epsilon dy \right| \leq C\epsilon^{\frac{3}{2}} (\epsilon^{m_j-1} + |y_{\epsilon,j} - y_j^0|^{m_j-1})\|\omega_\epsilon\|_\epsilon,$$

which implies that from Proposition 2

$$\left| \int_{B_d(y_\epsilon)} \frac{\partial Q}{\partial y_j} U_{y^0}^2\left(\frac{y-y_\epsilon}{\epsilon}\right) dy \right| = O(\epsilon^3 (\epsilon^{m_j + \min\{m_1, m_2, m_3\} - 1} + \max_{j=1,2,3} |y_{\epsilon,j} - y_j^0|^{2m_j - 1})). \tag{13}$$

On the other hand, we also find

$$\text{LHS of (13)} = c_j m_j \epsilon^3 \int_{B_{\frac{d}{\epsilon}}(0)} |\epsilon y_j + y_{\epsilon,j} - y_j^0|^{m_j - 2} (\epsilon y_j + y_{\epsilon,j} - y_j^0) U_{y^0}^2(y) dy$$
$$+ O(\epsilon^3 (\epsilon^m + |y_\epsilon - y^0|^m)). \tag{14}$$

Thus, (13) and (14) imply that

$$\int_{B_{\frac{d}{\epsilon}}(0)} |y_j + \frac{y_{\epsilon,j} - y_j^0}{\epsilon}|^{m_j - 2} (y_j + \frac{y_{\epsilon,j} - y_j^0}{\epsilon}) U_{y^0}^2(y) dy = O(\epsilon) + O(\epsilon^{m_j - 1}), \tag{15}$$

which, together with Proposition 2, gives that for $j = 1, 2, 3$,

$$|y_{\epsilon,j} - y_j^0| = O(\epsilon) \quad \text{and} \quad \|\omega_\epsilon\|_\epsilon = O(\epsilon^{\frac{3}{2} + \min\{m_1, m_2, m_3\}}).$$ (16)

Up to a subsequence, we can suppose that $\frac{|y_{\epsilon,j} - y_j^0|}{\epsilon} \to \tilde{y}$. Then, letting $\epsilon \to 0$ in (15), we have

$$\int_{\mathbb{R}^3} |y_j + \tilde{y}_j|^{m_j - 2}(y_j + \tilde{y}_j) U_{y^0}^2(y) dy = 0.$$

This gives that $\tilde{y} = 0$ since $U_{y^0}(|y|)$ is strictly decreasing with respect to $|y|$. So, $|y_\epsilon - y^0| = o(\epsilon)$. $\square$

### 3. Proof of the Main Theorem

Suppose that $u_\epsilon^{(j)}, j = 1, 2$ are two distinct solutions derived as in Proposition 1. By (11), $u_\epsilon^{(j)}, j = 1, 2$ are bounded functions in $\mathbb{R}^3$. Set

$$\eta_\epsilon = \frac{u_\epsilon^{(1)} - u_\epsilon^{(2)}}{\|u_\epsilon^1 - u_\epsilon^2\|_{L^\infty(\mathbb{R}^3)}}.$$

Then, $\|\eta_\epsilon\|_{L^\infty(\mathbb{R}^3)} = 1$, and similar to Propositions 6.1 and 6.2 in [5], we have

**Lemma 2.** *There holds*

$$\|\eta_\epsilon\|_\epsilon = O(\epsilon^{\frac{3}{2}}).$$

**Lemma 3.** *Letting $\bar{\eta}_\epsilon = \eta_\epsilon(\epsilon y + y_\epsilon^{(1)})$, then there exist $\beta_j \in \mathbb{R}$, $j = 1, 2, 3$ such that, up to a subsequence if necessary, $\bar{\eta}_\epsilon(y) \to \sum_{j=1}^3 \beta_j \frac{\partial U_{y^0}(y)}{\partial y_j}$ uniformly in $C^1(B_R(0))$ for any $R > 0$.*

**Lemma 4.** *Let $\beta_j$ be as in Lemma 3. Then, $\beta_j = 0$, for $j = 1, 2, 3$.*

**Proof.** Since $u_\epsilon^{(1)}, u_\epsilon^{(2)}$ are the positive solutions of (2), the Pohazaev identity (12) gives that

$$\int_{B_d(y_\epsilon^1)} \frac{\partial Q}{\partial y_j}(u_\epsilon^{(1)}(y) + u_\epsilon^{(2)}(y))\eta_\epsilon(y) dy = O(e^{-\frac{\sigma}{\epsilon}}).$$ (17)

On the other hand,

$$\int_{B_d(y_\epsilon^{(1)})} \frac{\partial Q}{\partial y_j}(u_\epsilon^{(1)}(y) + u_\epsilon^{(2)}(y))\eta_\epsilon(y) dy$$
$$= m_j c_j \int_{B_d(y_\epsilon^{(1)})} |y_j - y_j^0|^{m_j - 2}(y_j - y_j^0)(u_\epsilon^{(1)}(y) + u_\epsilon^{(2)}(y))\eta_\epsilon(y) dy$$ (18)
$$+ O\left(\int_{B_d(y_\epsilon^{(1)})} |y - y^0|^m (u_\epsilon^{(1)}(y) + u_\epsilon^{(2)}(y))\eta_\epsilon(y) dy\right).$$

Note that

$$u_\epsilon^{(1)}(y) + u_\epsilon^{(2)}(y) = 2U_{y^0}\left(\frac{y - y_\epsilon^{(1)}}{\epsilon}\right) + o(1)\nabla U_{y^0}\left(\frac{y - y_\epsilon^{(1)}}{\epsilon}\right) + O\left(\sum_{j=1}^2 |\omega_\epsilon^{(j)}|\right).$$ (19)

Then, it holds

$$\int_{B_d(y_\epsilon^{(1)})} |y_j - y_j^0|^{m_j-2}(y_j - y_j^0)(u_\epsilon^{(1)}(y) + u_\epsilon^{(2)}(y))\eta_\epsilon(y)dy$$

$$= 2\int_{B_d(y_\epsilon^{(1)})} |y_j - y_j^0|^{m_j-2}(y_j - y_j^0)U_{y^0}\left(\frac{y - y_\epsilon^{(1)}}{\epsilon}\right)\eta_\epsilon(y)dy \tag{20}$$

$$+ o(1)\int_{B_d(y_\epsilon^{(1)})} |y_j - y_j^0|^{m_j-2}(y_j - y_j^0)\nabla U_{y^0}\left(\frac{y - y_\epsilon^{(1)}}{\epsilon}\right)\eta_\epsilon(y)dy$$

$$+ O\left(\int_{B_d(y_\epsilon^1)} |y_j - y_j^0|^{m_j-1}(|\omega_\epsilon^{(1)}(y)| + |\omega_\epsilon^{(2)}(y)|)\eta_\epsilon(y)dy\right).$$

Now, since $\frac{\partial U_{y^0}(y)}{\partial y_j}$ is an odd function with respect to $y_j$ and an even function with respect to $y_i$ for $i \neq j$, using Lemma 3, we deduce that

$$\int_{B_d(y_\epsilon^{(1)})} |y_j - y_j^0|^{m_j-2}(y_j - y_j^0)U_{y^0}\left(\frac{y - y_\epsilon^{(1)}}{\epsilon}\right)\eta_\epsilon(y)dy$$

$$= \epsilon^{m_j+2}\int_{B_{\frac{d}{\epsilon}}(0)} \left|y_j + \frac{y_{\epsilon,j}^{(1)} - y_j^0}{\epsilon}\right|^{m_j-2}\left(y_j + \frac{y_{\epsilon,j}^{(1)} - y_j^0}{\epsilon}\right)U_{y^0}(y)\left(\sum_{j=1}^3 \beta_j\frac{\partial U_{y^0}(y)}{\partial y_j} + o(1)\right)dy \tag{21}$$

$$= \beta_j\epsilon^{m_j+2}\int_{\mathbb{R}^3} |y_j|^{m_j-2}y_j U_{y^0}(y)\frac{\partial U_{y^0}(y)}{\partial y_j}dy + o(\epsilon^{m_j+2}),$$

and similarly,

$$\int_{B_d(y_\epsilon^{(1)})} |y_j - y_j^0|^{m_j-2}(y_j - y_j^0)\nabla U_{y^0}\left(\frac{y - y_\epsilon^{(1)}}{\epsilon}\right)\eta_\epsilon(y)dy = O(\epsilon^{m_j+2}). \tag{22}$$

Also, Proposition 4 gives that

$$\int_{B_d(y_\epsilon^{(1)})} |y_j - y_j^0|^{m_j-1}(|\omega_\epsilon^{(1)}(y)| + |\omega_\epsilon^{(2)}(y)|)\eta_\epsilon(y)dy$$

$$= O(\epsilon^{m_j-1}(\|\omega_\epsilon^{(1)}\|_\epsilon + \|\omega_\epsilon^{(2)}\|_\epsilon)\|\eta_\epsilon\|_\epsilon) \tag{23}$$

$$= O(\epsilon^{m_j+2+\min\{m_1,m_2,m_3\}}).$$

Moreover, with the same argument, we obtain

$$\int_{B_d(y_\epsilon^{(1)})} |y - y^0|^m(u_\epsilon^{(1)}(y) + u_\epsilon^{(2)}(y))\eta_\epsilon(y)dy = O(\epsilon^{m+3}). \tag{24}$$

Then, from (20)–(24), it holds

$$\text{LHS of (18)} = 2c_j m_j\beta_j\epsilon^{m_j+2}\int_{\mathbb{R}^3} |y_j|^{m_j-2}y_j U_{y^0}(y)\frac{\partial U_{y^0}(y)}{\partial y_j}dy + o(\epsilon^{m_j+2}). \tag{25}$$

Thus, (17) and (25) imply that $\beta_j = 0$. $\square$

**Proof of Theorem 2.** Suppose that $u_\epsilon^{(j)}, j = 1, 2$ are two distinct solutions derived as in Proposition 1; then, $\|\eta_\epsilon\|_{L^\infty(\mathbb{R}^3)} = 1$ by assumption. But it follows from Lemmas 3 and 4 and the maximum principle that $\eta_\epsilon(y) = 0$. We reach a contradiction by constructing $\eta_\epsilon$. We find $u_\epsilon^{(1)} = u_\epsilon^{(1)}$, which proves that problem (2) has only one positive single-peak solution if $\epsilon$ is small enough. $\square$

**Author Contributions:** Methodology, S.Y.; Formal analysis, B.S.; Writing—original draft, J.L. All authors have read and agreed to the published version of the manuscript.

**Funding:** This research received no external funding.

**Data Availability Statement:** The data will be made available by the authors on request.

**Conflicts of Interest:** Shichao Yi was employed by Yangzijiang Shipbuilding Group. The remaining authors declare that the research was conducted in the absence of any commercial or financial relationships that could be construed as a potential conflict of interest. The Yangzijiang Shipbuilding Group had no role in the design of the study; in the collection, analyses, or interpretation of data; in the writing of the manuscript, or in the decision to publish the results.

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
