# Peer review of "Uniqueness of Single Peak Solutions for a Kirchhoff Equation"

_mathematics, doi:10.3390/math12101462_

Round 1

Reviewer 1 Report

Comments and Suggestions for Authors

Kirchhoff-type equations are common in mathematical physics and can take on various forms. Proving the uniqueness of single peak solutions for a Kirchhoff equation typically requires a combination of analytical techniques, such as variational methods, maximum principles, and Pohozaev identities, depending on the specific equation and conditions. Finding a solution to this problem can be challenging and may not always have a straightforward answer. The authors discuss the distinctiveness of the solution to a singular perturbation Kirchhoff equation, which may rely on additional assumptions or restrictions imposed on the problem. This differential equation includes a small parameter, known as the perturbation parameter, multiplying a term that results in singular behavior. This equation arises in various fields of physics and engineering, including mechanics, fluid dynamics, and mathematical biology. The author should conduct a comprehensive literature review to provide context and support for the problem being studied. The paper concludes with the proof of Theorem 1.1, which is the main objective of the paper. Furthermore, a more structured conclusion, as is typical in academic papers, would enhance the paper's overall organization.

Besides, the paper begins with stating the problem and finishes with its proof. Given my background as a mechanical engineer who delves into mathematical issues due to the importance of mathematical knowledge for mechanical problems, there may be slight disparities in how we articulate the problem compared to pure mathematicians. Consequently, I anticipated the paper to commence with a clear motivation and justification for investigating the problem at hand. Furthermore, it would have been beneficial for the paper to conclude by emphasizing the significance of the findings, aspects that are currently unfortunately absent.

Comments on the Quality of English Language

-

Author Response

Dear Reviewer Many thanks for your comments and advices. After carefully studying the comments and advices, we have made corresponding changes to the paper. In the introduction, we introduce the kirchhoff equation from the wave equation, which makes the study equations have a stronger application context. The modifications are marked in red. If you need any other information, please do not hesitate to let us know. Yours sincerely

Reviewer 2 Report

Comments and Suggestions for Authors

In this article author prove the uniqueness of the concentrated solutions under some suitable assumptions on asymptotic behaviors of Q(y) and its first derivatives by using a type of Pohozaev identity for $\epsilon$ small enough. The results are interesting. But the is not presented properly. 
Comment-1: What is intension behind the value of $1<p<5$? Where did you use it?

Comment-2: Add the MSC-2021.
Comment-3: What is the meaning $Q_3$ bar? You did not mention anywhere! 
Comment4: On page 3 the equation is going from outside the page. Arrange that equation.
Comment 5: In proposition 2.4, what is the meaning of notation $R>>1? If you are taking R is very larger than 1 then write in language or mention the meaning the symbol meaning.

Comment 6: Conclusion is not included. Conclude your work in last section of the article.

Comment 7: Modify the introduction.
After the above-mentioned modifications, I will recommend for publication in Mathematics.

Author Response

Dear Reviewer,

  Many thanks for your comments and advices. After carefully studying the comments and advices, we have made corresponding changes to the paper in red. If you need any other information, please do not hesitate to let us know.

Yours sincerely

Response to Reviewer:

  1. What is intension behind the value of $1<p<5$? Where did you use it?

Answer: In this paper, $1<p<5$ means the exponent p is subcritical and for example we use it in (2.7) where we use the continuous embedding of H^1(R^3) to L^{p+1}(R^3)

  1. Add the MSC-2021.

Answer: Thanks for the reminder.  MSC:35A01; 35A02; 35B25; 35J20; 35J60.

  1. What is the meaning $Q_3$ bar? You did not mention anywhere!

Answer: In this paper, it just is a assumption on Q(y) and we use it to get (2.12) and (3.12) in Proposition 2.5 and Lemma 3.3 respectively

  1. On page 3 the equation is going from outside the page. Arrange that equation.

Answer: Thank you for your advice. We arranged the equation.  

  1. In proposition 2.4, what is the meaning of notation $R>>1? If you are taking R is very larger than 1 then write in language or mention the meaning the symbol meaning.

Answer: This means R is very large than 1. We can change it to “R large enough”.

  1. Conclusion is not included. Conclude your work in last section of the article.

Answer:  Our conclusion is that by constructing $\eta_\epsilon$ and getting a contradiction, we find $u^{(1)}_\epsilon= u^{(1)}_\epsilon$, which proves that problem (1.2) has only one positive single-peak solution if $\epsilon$ small enough

  1. Modify the introduction.

Answer: Thank you for your advice. We have reworked the introduction and added a paragraph on the history of the development of equations.

Round 2

Reviewer 2 Report

Comments and Suggestions for Authors

Authors made all necessary modifications. I recommend this article for publications in MATHEMATICS.